# The Evaluation of Facial Muscles by Surface Electromyography in Very Preterm Infants

**DOI:** 10.3390/biomedicines10112921

**Published:** 2022-11-14

**Authors:** Oskar Komisarek, Roksana Malak, Jacek Kwiatkowski, Katarzyna Wiecheć, Tomasz Szczapa, Joanna Kasperkowicz, Maja Matthews-Kozanecka, Teresa Matthews-Brzozowska, Małgorzata Wójcik, Włodzimierz Samborski, Ewa Mojs

**Affiliations:** 1Department of Plastic, Reconstructive and Aesthetic Surgery, Collegium Medicum in Bydgoszcz, Nicolaus Copernicus University in Torun, 85-821 Bydgoszcz, Poland; 2Department and Clinic of Rheumatology, Rehabilitation and Internal Medicine, Poznań University of Medical Sciences, 61-545 Poznań, Poland; 3Students Scientific Society of Maxillofacial Orthopaedics and Orthodontics, University of Medical Sciences, 60-812 Poznań, Poland; 4Department of Clinical Psychology, Poznań University of Medical Sciences, 60-812 Poznań, Poland; 5Neonatal Biophysical Monitoring and Cardiopulmonary Therapies Research Unit, II Department of Neonatology, Poznan University of Medical Sciences, 60-535 Poznan, Poland; 6Department of Social Sciences and the Humanities, Poznan University of Medical Sciences, 60-806 Poznań, Poland; 7Department of Orthodontics and Masticatory System Dysfunction, Poznan University of Medical Sciences, 60-812 Poznań, Poland; 8Department of Physiotherapy, Faculty of Physical Culture in Gorzów Wielkopolski, Poznan University of Physical Education, Estkowskiego 13, 66-400 Gorzów Wielkopolski, Poland

**Keywords:** sEMG, surface electromyography, masticatory muscles, facial muscles, preterm infant, neurodevelopment, early intervention

## Abstract

Background: It is reported that 40% of preterm infants have problems with eating. Neonatal feeding disorders may be one of the factors increasing neonatal mortality. The aim of our study was to evaluate the muscles involved in suckling and swallowing in premature newborns using surface electromyography (sEMG). We would like to objectively describe the tension of muscles engaged in feeding in order to properly plan the therapy. Another aim was to compare sEMG measurements to gestational age, birth weight, and umbilical blood pH to show which parameters put children at risk of feeding problems. Methods: Sixteen preterm neonates with gestational age less than 32 weeks, birth weight less than 1500 g, and oral feeding difficulties were analyzed for muscle response and electrical activity of nerves using sEMG (surface electromyography). Results: We found a negative correlation indicating that preterm infants with a younger gestational age had higher suprahyoid muscle tension, and a positive correlation was found between pH value and suprahyoid muscles. The lower the pH value, the lower the tension in the suprahyoid muscles. Conclusions: sEMG may be a helpful diagnostic tool in the evaluation of the masticatory system of premature infants. Due to the abnormal tone of the muscles responsible for swallowing, it is advisable to rehabilitate as early as possible.

## 1. Introduction

It is reported that 40% of preterm infants have problems with eating. Preterm birth is one of the most common causes of infant mortality, and it is associated with many complications, including underdeveloped lungs, a deformed cardiovascular system, immature neural functions, a weakened immune system, an abnormal gastrointestinal tract, and feeding intolerance. Due to many life-threatening complications and underdeveloped organs, preterm neonates are frequently intubated and fed through nasogastric or orogastric tubes for prolonged periods, leading to muscle incompetency and poor feeding outcomes [1]. Neonatal feeding disorders may be one of the factors increasing neonatal mortality [2]. Effective and efficient feeding allows premature newborns to develop properly in terms of emotions, socialization and staying in good health [3]. It is essential to understand the functions of facial expressions and chewing muscles during suckling and swallowing to obtain information on the development of the stomatognathic system in a premature newborn. This allows one to assess the possibility of chewing disorders, speech articulation, swallowing, breathing, and malocclusion [4]. An electromyographic examination (EMG) is a basic type of examination of electro-neurophysiology, which is the determination of the electrical activity of the muscles. The electromyographic examination allows for quick and repetitive assessments of the functional characteristics of a muscle and even multiple muscles simultaneously. There are two types of electromyographic examination: quantitative with an electrode needle, which is considered the gold standard in the assessment of motor activity of individual motor muscles, and global electromyography using electrode surface electromyography (sEMG).This examination enables an objective and qualitative assessment of muscle work. Surface electromyography, due to the lack of discomfort that arises when inserting needles into the muscles in the needle method, has an advantage in the head and neck [5]. This study aimed to evaluate the muscles involved in suckling and swallowing in premature newborns using surface electromyography.

## 2. Materials and Methods

This study was approved by the Bioethics Committee, consent ref. No. 481/21.

Sixteen preterm neonates were hospitalized at the Gynecology and Obstetrics Clinical Hospital at Poznan University of Medical Sciences. All 16 participants were of Caucasian descent and presented feeding problems. Inclusion criteria included: preterm neonate with gestational age of less than 32 weeks; birth weight of less than 1500 g, and oral feeding difficulties. All of the patients had feeding difficulties and were tube-feeding. Exclusion criteria: neonate with oxygen desaturation (SpO_2_ < 88%), heart rate slower than 100 or faster than 205 beats per minute, hypotension, active inflammation, bone malformation replacement, tumors, encephalopathy, or high-mortality congenital anomalies (e.g., Edwards’ syndrome, Patau’s syndrome). The average gestational age was 29 ± 2 (minimum 25 weeks postmenstrual and maximum 32 weeks postmenstrual). The average birthweight was 1230 ± 229 g.

Each patient’s muscle electrical activity was analyzed using surface electromyography (sEMG); model Noraxon MR3 version 3.12.70 with EMG TYCO/Kendall H124SG ECG pediatric electrodes, Ø 24 mm (disposable, self-adhesive silver chloride electrodes). All measurements were taken in a quiet and isolated room to restrict environmental influences on the subjects. Infants’ skin was pre-wiped with sterile compresses soaked in an alcohol solution to limit electrode conductivity disturbances. The electrodes were placed on the lower border of the orbicularis oris muscle, masseter muscles, temporal muscles, and suprahyoid muscles on both the left and right sides of the patients (Figure 1). The obtained sEMG signal was conditioned, quantified and standardized according to Boxtel [6]. Sucking reflexes were stimulated by finger covered by nitrile gloves inserted into the infants’ oral cavities. All measurements were performed while suckling. We decided to use non-nutritive suckling because it was shown in many studies to be a safe and adequate way of not only checking but also promoting feeding performance [7]. We did not use boluses in order to avoid aspiration. The maximum voluntary contraction (MVC) was measured for every muscle group, then results were recorded and compared to gestational age, birth weight, and umbilical blood pH. The aim of our study was to evaluate the muscles involved in suckling and swallowing in premature newborns using surface electromyography (sEMG). We would like to objectively describe the tension of muscles engaged in feeding in order to properly plan therapy in the future. Another aim was to compare sEMG measurements to gestational age, birth weight, and umbilical blood pH in order to show which parameters place preterm infants at risk of feeding problems.

Results taken from sEMG were statistically analyzed using Statistica version 13 software (TIBCO Software, Tulsa, OK, USA). Results were considered statistically significant when *p* < α, α = 0.05. The measurement data were also examined with the Shapiro–Wilk normality test to determine if they were well-modeled by a normal distribution. To compare the variables in the study group, in the case of compliance with the normal distribution and equal variances, the Student’s t-test was used for samples. When variance was unequal, we used the Cochran–Cox test, and when there was no compliance with the normal distribution, the Mann–Whitney test. To investigate the relationship between the variables, Pearson’s r coefficient of linear correlation was calculated in the case of compliance with the normal distribution; in case of disagreement with the normal distribution or for ordinal variables, Spearman’s rank correlation coefficient, Rs, was calculated.

## 3. Results

The descriptive statistical results from our study are presented in Table 1.

The average gestational age was 29 ± 2 (minimum 25 weeks postmenstrual and maximum 32 weeks postmenstrual). The average birth weight was 1230 ± 229 g.

Week of gestation and pH were compared to the left and right suprahyoid muscles during maximum voluntary contraction (MVC). As shown in Table 2, we found a negative correlation indicating that preterm infants with a younger gestational age had higher suprahyoid muscle tension.

A positive correlation was found between pH value and the right and left suprahyoid muscles during MVC. The lower the pH value, the lower the tension in the suprahyoid muscles (Table 3). We found no statistically significant difference in other muscles (orbicularis oris muscles, masseter muscles, temporal muscles).

## 4. Discussion

Surface electromyography (sEMG) is a useful and non-invasive method for the early detection of orofacial muscle hypotonicity in preterm neonates [8].The studies by Greenfield et al. from 1958 and Brooke et al. from 1979 suggest that EMG should not be performed in the first 3 months of life because it is hard to interpret, and results could be of no value [9,10]. Nevertheless Packer R.J., in the study from 1982, confirmed that infant sEMG measurements are valuable, in line with the sensitivity of the sEMG examination in adults [11]. sEMG, in our opinion, has some limitations. For example, if an infant’s skin has been oiled with creams, it can cause problems when applying the sEMG electrodes even if an alcohol solution is used to pre-wipe the skin. Additionally physiological functions such as crying and head and arm movements are causes of artifacts throughout the examination of muscle electrical activity. These factors require repeating the sEMG examination and carefully removing false results from the obtained data. In our study, we examined the suprahyoid muscles, lower pole of orbicularis oris, and messenger and temporalis muscles on the left and right sides of the head. We could not examine a single muscle, which is another limitation of sEMG [12]. There is an option to take advantage of the needle EMG method to examine every single suprahyoid muscle, but it is painful and uncomfortable [13], and its use, therefore, is firmly limited in neonatology. sEMG may be a good additional diagnostic tool for the physiotherapist, but requires special training, and the person who conducts the examination should be aware of the limitations of this examination method.

Collected statistical data had shown that preterm infants with a younger gestational age had lower suprahyoid muscle tension. Our study reported that preterm infants need early oral stimulation therapies to provide proper nutrition and develop correct coordination of suckling, swallowing and breathing in hypotonic muscles.

Hypotonia may be the result of a variety of factors. The most common cause is a central disorder in the brain or spinal cord, but other causes may be located in the peripheral nervous system; therefore, the reason for hypotonia may be located in every part of the motor unit. It could be caused by genetic/congenital disorders, for example, in spinal muscular atrophy, congenital myasthenia, or acquired, for example, in poliomyelitis or Guillain–Barre syndrome [14]. Another reason for hypotonia may be low umbilical arterial pH [15]. We showed in our study that children with low values of umbilical arterial pH who are at risk of hypoxia and encephalopathy have lower muscle tone in the suprahyoid muscles, and children with higher umbilical arterial pH have higher muscle tone in the suprahyoid muscles.

Marsha Walker in the study from 2008 notes that some late preterm infants can suckle the breast during feeding and swallow, especially with the support of the jaw. Other newborns are not able to suckle properly because they do not have enough strength, or they quickly lose adequate force to hold the nipple into the mouth and suckle [16]. McBride, M. C. and Danner, S. C suggest a “dancer hand”, where using the forefinger and thumb to form a U-shaped hand around the newborn’s mouth helps hypotonic infants in nutrition because, apart from supporting the jaw, the mother’s hand supports also the cheeks and provides more pressure [17]. The study by L.C. Gilstrap et al. on 2738 term newborns confirmed that umbilical blood with pH of less than <7 with an Apgar score ≤3 at 1 min is correlated with hypotonia, but also with seizures [18].

Infants born preterm are at high risk of oral feeding difficulties; to have safe neonate oral feeding, it is critical to have healthy and competent organ systems with satisfactory muscle coordination of sucking, swallowing, and breathing [19]. Infants without feeding competency will either be required to be on intravenous nutritional support or nasogastric or orogastric tube intubation until their sucking, swallowing, and breathing coordination is established. Adequate nutritional supply is critical to preterm neonate survival, so it is important to start intervention early. Dodrill et al.’s study found that prolonged periods of enteral feeding were correlated with altered oral sensitivity and caused muscle weakness and delayed motor skill developmental in preterm neonates [20]. Therefore, it is important to identify the true underlying cause of neonate oral feeding difficulty and choose the appropriate therapy. If the reason for oral feeding difficulties is neuronal/behavioral/environmental, it will have to be determined and corrected, whereas a preterm neonate with oral feeding difficulties because of muscle hypotonicity needs nutritional support. That is why he or she will have to be supplemented with oral stimulation therapies to prevent further delays in the motor coordination needed for oral feeding tolerance.

We found that the earlier the week of gestation, the higher the tonus in the suprahyoid muscles. Similar results were reported by Cameron [21], who presented results contrary to those of other researchers, namely, that if the week of gestation was earlier, then children presented higher muscle tone. Maybe the reason is that the muscles of the head are rigidly maintained [22]. Another reason could be an imbalance in muscle tone in children born before the 33rd week of gestation [23]. Preterm infants were also more hyperexcitable and tended to present imbalance in muscle tone [24]. That is why children born before the 33rd week of gestation did not present head lag. Head lag means that the infant’s head seems to flop around or lags posteriorly behind the trunk during the pull-to-sit maneuver [25]. On the other hand, the birth before the 32nd week of gestation with a birth weight of less than 1500 g, hypoxic-ischemic brain damage, birth asphyxia, and dysfunctional pregnancy (infectious or somatic pathology) put newborns at risk of symptoms that lead to spastic diplegia, meaning that they present muscle hypertonia [26].

In another study, preterm infants presented higher activity of the suprahyoid musculature during feeding. This may be explained by the greater range of tongue movement, as premature infants usually perform tongue protrusion to get the milk [26,27].

Additionally, the period for checking the muscles of the head may be too early. Poor ability to maintain balance in the muscles of the head is typical in preterm children. The balance in muscle tone of the neck, head and trunk should appear during the pull-to-sit task at 3–4 months corrected age, and then head lag can be associated with poorer future motor outcome scores [23]. These findings highlight the role of very early pull-to-sit skills for later social and language outcomes. Atypical postural development and persistent presence of head lag may be important early indicators of social and language vulnerabilities, including autism spectrum disorder (ASD). Only 16 preterm neonates suffering from oral difficulties were examined. For a statistical study, a larger sample size will increase the significance and confidence of the findings. With surface EMG, we could not examine single muscles; therefore, the results were from the groups of muscles located under the electrodes. sEMG has been proven useful in determining muscle hypotonicity in oral-feeding intolerant neonates, but further research monitoring the progress and obtaining 5-year follow-up measurements with sEMG would be useful in determining the stability and long-term prognosis of oral stimulation rehabilitation.

To summarize, we demonstrated that sEMG is a valuable method of examining infants who are at risk of feeding problems, as other studies also found [28,29,30].

## 5. Conclusions

sEMG may be a helpful diagnostic tool in the evaluation of the masticatory system of premature infants. Due to the abnormal tone of the muscles responsible for swallowing, it is advisable to rehabilitate as early as possible.

## Figures and Tables

**Figure 1 biomedicines-10-02921-f001:**
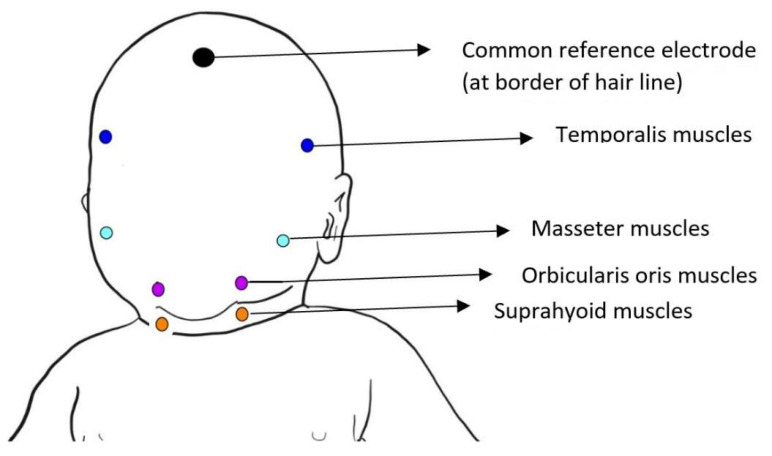
Electrode locations for measuring orofacial EMG activity.

**Table 1 biomedicines-10-02921-t001:** Descriptive statistics of test subjects, with neonates’ gestational age, birth weight, Apgar score, and blood pH at the time of birth.

Variable	N	Median	Minimum	Maximum
Gestational age (weeks)	16	29	25	32
Birth weight (g)	16	1305.00	725.00	1520.00
Apgar score	16	8	5	9
pH	15	7.33	6.96	7.40

**Table 2 biomedicines-10-02921-t002:** The relationship between gestational age and maximum voluntary contraction (MVC) of left and right suprahyoid muscles.

Variable	N	R Spearman	*p*
Week of gestation and sEMG MVC left suprahyoid	16	−0.542	0.03
Week of gestation and sEMG MVC right suprahyoid	16	−0.669	0.005

**Table 3 biomedicines-10-02921-t003:** The relationship between pH and maximum voluntary contraction (MVC) of left and right suprahyoid muscles.

Variable	N	R Spearman	*p*
pH and sEMG MVC left suprahyoid	15	0.691	0.004
pH and sEMG MVC right suprahyoid	15	0.677	0.006

## Data Availability

The data that support the findings of this study are available per request from the corresponding authors (O.K. and R.M.).

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
