# Peer review of "The Evaluation of Facial Muscles by Surface Electromyography in Very Preterm Infants"

_biomedicines, 2022, doi:10.3390/biomedicines10112921_

Round 1
Reviewer 1 Report
Dear Authors,
This is a very interesting manuscript about the muscles involved in suckling and swallowing in premature newborns using surface electromyography. I have some recommendations and comments.
Title - check the correct spelling ("electromography")
Abstract - line 29 - you allready explained what sEMG means (this is redundant).
Materials and Methods
-Please define better the aim of the study and what you expect to found (Primary and secondary outcomes).
Line 77- check - The average "Birth weight"
Lines 75-77 - it would fit better in the Results chapter
Results
Table 2, 3 - use only three decimals for p and R
- Explain the abbreviations under the tables
Discussion
- In a sentence or two at the beginning of the Discussion chapter, please summarize the main findings of your study, and then compare with similar studies
Line 133 - "messenger muscle"?
Line 180 - please check "10.1016/j.earlhumdev.2015.01.001."
Author Response
Dear Editors and Reviewers,
Thank You for all valuable suggestions, for Your time and attention. We are honored to have the chance to publishing the article in Biomedicines
Of course, we agree that we should check the title - the correct spelling ("electromography") and we corrected into “electromyography”
The Reviewer 1st wrote "Abstract - line 29 - you allready explained what sEMG means (this is redundant).” That is why we removed it from the line 30 - 31 st
The Reviewer 1st wrote:
“Please define better the aim of the study and what you expect to found (Primary and secondary outcomes).”
That is why we put the aim in the line 26 – 29 and 94 - 100
The Reviewer 1st wrote: “Line 77- check - The average "Birth weight"”. We added the word “weight”.
The Reviewer 1st wrote: “Lines 75-77 - it would fit better in the Results chapter”. That is why we put the line 75 – 77 into the results.
The Reviewer 1st wrote: that in the results in the Table 2, 3 we should use only three decimals for p and R. We have done it.
The Reviewer 1 st wrote: Explain the abbreviations under the tables. We have put the explanation under the tables
The Reviewer 1st wrote: In a sentence or two at the beginning of the Discussion chapter, please summarize the main findings of your study, and then compare with similar studies. We add a comparsion in line 176-180.
The Reviewer 1st wrote: Line 133 - "messenger muscle"?. Of course, we agree that it should be masseter muscle.
The Reviewer 1st wrote: Line 180 - please check "10.1016/j.earlhumdev.2015.01.001." Reference have been corrected
Reviewer 2 Report
I have read this paper with interest, but do have several - in my opinion - relevant comments and concerns. I have provided these comments consecutively, not necessarly reflecting their relevance.
the title suggest facial muscles, while the muscles involved in suckling and swallowing were assessed.
it is completely unclear how the 16 cases were selected, and how has 'oral feeding difficulties' been assessed/quantified. There are scores to do so ?
A picture of preterm or drawing with the electrodes might be useful so better understand the approach taken.
The same holds true for the 'signal' as documented ? how do you qantify this, and does this truly reflect tension, or could this also be affected by eg subcutis and cutis, or the size of the muscle ?
The last sentence of the abstract is not supported by the data collected and reported.
the study describes suckling and swallowing, but these are 'dry swallow' without boluses. This is another major limitation in my assessment of the paper.
The suface EMG is suggested to be objective and qualitative, but how does this work (cfr comments higher: truly tonus, or muscle size and subcutaneous anatomy ?)
what do you mean with 'bone replacement' as a contraindication ?
postmenstrual age is likely gestational age ? (line 76)
how have the sEMG data been linked to the sucking ? if sucking is more frequent or less frequent, how does this affect sEMG ?
What do you mean with compare variables between 2 groups (i cannot find such analysis)
tension is used as a synonym of sEMG signals ? is this correct ? (throughout the paper). I have not read robust arguments to do so.
why do you mention 'electrical activity of nerves'
messenger ? should this read masseter ?
a best this is a pilot study, reporting on the feasibility to consider sEMG as a research tool, and these data should not be overinterpreted
Author Response
Thank You for all valuable suggestions, for Your time and attention. We are honored to have
the chance to publishing the article in Biomedicines
The Reviewer 2nd wrote that “the title suggest facial muscles, while the muscles involved in suckling
and swallowing were assessed”, so we have changed it into orofacial muscles. Their main functions
are chewing, swallowing, and speech.
The Reviewer 2nd wrote that ,,it is completely unclear how the 16 cases were selected, and how has
'oral feeding difficulties' been assessed/quantified. There are scores to do so ? “ We selected to the
study patients with feeding difficulties, who were tube feeding and who were born under 32 week of
gestation and whose birth weight were below 1500 g. line 74
The Reviewer 2nd wrote “A picture of preterm or drawing with the electrodes might be useful so better
understand the approach taken.” That is why we added the figure 1. Line 143
The Reviewer 2nd wrote “The same holds true for the 'signal' as documented ? how do you qantify this,
and does this truly reflect tension, or could this also be affected by eg subcutis and cutis, or the size of
the muscle ? The suface EMG is suggested to be objective and qualitative, but how does this work (cfr
comments higher: truly tonus, or muscle size and subcutaneous anatomy ?)
The obtained sEMG signal was conditioned, Quantificated and standardized according to Boxtel.van
Boxtel, A. (2010). Facial EMG as a tool for inferring affective states. In A. J. Spink, F. Grieco, O. Krips,
L. Loijens, L. Noldus, & P. Zimmerman (Eds.), Proceedings of Measuring Behavior 2010 (pp. 104-
108). Noldus Information technology. Line 88
The Reviewer 2nd wrote “The last sentence of the abstract is not supported by the data collected and
reported. We agree and removed it
The Reviewer 2nd wrote “the study describes suckling and swallowing, but these are 'dry swallow'
without boluses. This is another major limitation in my assessment of the paper. We decided to use
non nutritive sucking because it was shown in many studies as the way safe and adequate way of
checking feeding performance .https://pubmed.ncbi.nlm.nih.gov/33905427/ . We did not use boluses
in order to avoid aspiration.
The Reviewer 2nd is right that we should change the word 'bone replacement'. Therefore we changed
into bone malformation
The Reviewer 2nd asked : postmenstrual age is likely gestational age ? (line 76), Yes, The Reviewer
2
nd is right it is the same.
The Reviewer 2nd wrote ,,how have the sEMG data been linked to the sucking ? if sucking is more
frequent or less frequent, how does this affect sEMG ? “ We counted rhythmical acts of sucking which
was reflected in the picture of sEMG.
The Reviewer 2nd asked” What do you mean with compare variables between 2 groups (i cannot find
such analysis)”. The Reviewer 2nd is right that we compared variables in only 1 group ( very preterm
infants who were tube fed because of feeding difficulties).
The Reviewer 2nd wrote tension is used as a synonym of sEMG signals ? is this correct ? (throughout
the paper). I have not read robust arguments to do so.
Yes, The Reviewer 2nd is right. The amplitude in the sEMG signal after signal processing reflects the
muscle tone.
The Reviewer 2nd wrote ,,why do you mention 'electrical activity of nerves'” We removed it from text. In
the conducted examination, we cannot correctly assess the electrical activity of the nerve. Line 31,80
and 159,
The Reviewer 2nd wrote messenger should this read masseter ? Of course, we agree that it should be
massenter muscle. Line 133
The Reviewer 2nd wrote ,,a best this is a pilot study, reporting on the feasibility to consider sEMG as a
research tool, and these data should not be overinterpreted” We are totally agree with the comments
of the reviewer.
Round 2
Reviewer 1 Report
Dear authors,
The manuscript is improved.
Regards.
Author Response
Dear Editors and Reviewers,
Thank You for all valuable opinion as well as for Your time and attention.
We are honored to have the chance to publishing the article in Biomedicines
Your sincerely,
Oskar Komisarek
Reviewer 2 Report
i have read the revised version. it has some value as it describes the feasibility of the technique.
two suggestions: arteriar should read arterial
please reconsider the text flow of the two last alineas of the discussion. I do not really understand what you mean with 'we have found some limitations' and subsequent comment on the size of the study, as this is not a technical limitation but a study specific limitation. I highly recommend to reconsider this.
Author Response
Dear Editors and Reviewers,
Thank You for all valuable suggestion as well as for Your time and attention.
We are honored to have the chance to publishing the article in Biomedicines.
The Reviewer 2nd wrote: “two suggestions: arteriar should read arterial” – that is why we put
179, 180, 182.
Thank You, we have changed it.
The Reviewer 2nd wrote: “please reconsider the text flow of the two last alineas of the
discussion. I do not really understand what you mean with 'we have found some limitations'
and subsequent comment on the size of the study, as this is not a technical limitation but a
study specific limitation. I highly recommend to reconsider this”. That is why we have
changed the discussion in the line 231 – 254 in order to achieve a flow in last alineas.
Yours sincerely,
Oskar Komisarek